# Chemical, Bioactivity, and Biosynthetic Screening of Epiphytic Fungus *Zasmidium pseudotsugae*

**DOI:** 10.3390/molecules25102358

**Published:** 2020-05-19

**Authors:** Gisela A. González-Montiel, Elizabeth N. Kaweesa, Nicolas Feau, Richard C. Hamelin, Jeffrey K. Stone, Sandra Loesgen

**Affiliations:** 1Department of Chemistry, Oregon State University, Corvallis, OR 97331, USA; gonzalgi@oregonstate.edu (G.A.G.-M.); el.kaweesa@whitney.ufl.edu (E.N.K.); 2Whitney Laboratory for Marine Bioscience and Department of Chemistry, University of Florida, St. Augustine, FL 32080, USA; 3Department of Forest and Conservation Sciences, University of British Columbia, Vancouver, BC V6T 1ZA, Canada; nicolas.feau@ubc.ca; 4Faculté de Foresterie et Géomatique, Institut de Biologie Intégrative et des Systèmes (IBIS), Université Laval, Québec, QC G1V 0A6, Canada; Richard.Hamelin@ubc.ca; 5Department of Botany and Plant Pathology, Oregon State University, Corvallis, OR 97331, USA; stonej@science.oregonstate.edu

**Keywords:** epiphytic fungi, *Zasmidium pseudotsugae*, *Mycosphaerellaceae*, cytotoxic activity, quinone, 8,8′-bijuglone, genome mining

## Abstract

We report the first secondary metabolite, 8,8′-bijuglone, obtained from pure cultures of the slow growing Douglas fir- (*Pseudotsuga menziesii* var. *menziesii*) foliage-associated fungus *Zasmidium pseudotsugae*. The quinone was characterized using extensive LC/MS and NMR-based spectroscopic methods. 8,8′-Bijuglone exhibited moderate antibiotic activity against Gram-positive pathogens and weak cytotoxic activity in the NCI-60 cell line panel and in our in-house human colon carcinoma (HCT-116) cell line. An analysis of the fungal genome sequence to assess its metabolic potential was implemented using the bioinformatic tool antiSMASH. In total, 36 putative biosynthetic gene clusters were found with a majority encoding for polyketides (17), followed by non-ribosomal peptides (14), terpenes (2), ribosomal peptides (1), and compounds with mixed biosynthetic origin (2). This study demonstrates that foliage associated fungi of conifers produce antimicrobial metabolites and suggests this guild of fungi may present a rich source of novel molecules.

## 1. Introduction

The foliar fungal microbiome of Douglas-fir evergreen trees (*Pseudotsuga menziesii* var. *menziesii*) is a complex and diverse microbial ecosystem with underexplored chemical ecology [1,2]. Douglas-fir is an important commercial conifer species in forest plantations for timber and also most commonly used for Christmas trees in western North America, Europe, Australia, and New Zealand [3,4]. However, fungal pathogens, such as *Nothophaeocryptopus gaeumannii*, the causative agent of Swiss needle cast, have become a serious problem in Douglas-fir populations of western North America, where it has negatively affected the growth and production of this native conifer [2,4,5,6,7,8]. Plant-associated fungi often produce mycotoxins, or secrete effector proteins to suppress the immunity and defense mechanisms of their host [9,10,11], but the chemical interactions within fungal microbiome, ranging from mutualistic to pathogenic, are still elusive.

Herein, we focused on the Douglas-fir associated fungus *Zasmidium pseudotsugae* and its specialized metabolism. Both *Z. pseudotsugae* and *N. gaeumannii* are member of the *Dothideomycetes Capnodiales*, *Mycosphaerellaceae* [12] and frequently occur together in close association on Douglas-fir foliage. However, *Z. pseudotsugae* is a relatively benign epiphyte unlike *N. gaeumannii* which causes a foliage disease. While many natural products are known from Ascomycota, relatively few natural products are known from the *Mycosphaerellaceae* [12,13,14,15]. The best known are dothistromin, an aflatoxin related compound from *Dothistroma septosporum*, and metabolites from *Cercospora*, including perylquinones, for example the light activated phytotoxin cercosporin, and the beticolins (also called cebetins), which are polyketide derived toxins [16]. Therefore, in our continued approach to discover new chemical entities from different environments [17,18], plant-fungi communities have been shown to be an excellent source for new natural products [19,20,21,22].

Genome mining is now widely used for identification and connection of biosynthetic gene clusters (BGCs) found in microorganisms and plants to their respective specialized metabolites [23,24,25]. Bioinformatic tools have become an important complementary asset for compound-, or activity-driven natural product discovery. Here, we include the bioinformatics-based prediction of the chemical potential of *Zasmidium pseudotsugae* and surprisingly, 36 biosynthetic gene clusters were present in its genome. In solid-phase cultivation of the fungus, only one metabolite was produced, named 8,8′-bijuglone (**1**). The quinone was characterized by a combination of LC/MS and one- and two-dimensional NMR techniques and assessed for its antimicrobial and cytotoxicity activity. To our knowledge, this is the first bioactive metabolite isolated from *Z. pseudotsugae*.

## 2. Results and Discussion

### 2.1. Isolation and Characterization of 8,8′-Bijuglone (***1***)

*Zasmidium pseudotsugae* isolated from the needles of *P. menziesii* var. *menziesii* was grown on 2% malt-based agar for six months. The plate was then extracted with ethyl acetate and the agar culture extract was analyzed by LC/MS which revealed only one dominant metabolite eluting at a retention time of 21 min (Appendix A). Negative ionization mode gave an *m/z* value of 345.0 for [M − H]^−^ and 712.9 *m/z* for [2M − 2H + Na]^−^, while positive ionization mode detected an *m/z* value of 347.1 for [M + H]^+^. The compound was purified by semipreparative HPLC using an isocratic elution of 50% ACN/50% H_2_O both with 0.05% formic acid. The compound was identified as 8,8′-bijuglone (**1**) by 1D and 2D NMR (Table 1 and Appendix A) and referenced to existing NMR data from synthesized 8,8′-bijugone [26].

The ^13^C NMR spectrum displayed two quaternary carbons (δ_C_ 190.83 ppm and 184.89 ppm) corresponding to the two ketones, a phenol carbon (δ_C_ 161.91 ppm) corresponding to the carbon bearing the hydroxyl, three quaternary aromatic carbons, and four aryl hydrocarbons. The ^1^H NMR spectrum exhibited an intramolecular hydrogen bonding proton singlet (δ_H_ 12.49 ppm) and four aryl hydrocarbon doublets. Determination of the structure of **1** required 2D NMR correlation experiments including COSY, HSQC, and HMBC. Utilizing COSY, two distinct spin systems were observed corresponding to two separate sets of aryl hydrocarbons coupling to each other. The doublet of 3-H (δ_H_ 6.92 ppm) correlated with the doublet of 2-H (δ_H_ 6.71 ppm) and the doublet of 7-H (δ_H_ 7.24 ppm) correlated with the doublet of 6-H (δ_H_ 7.31 ppm). The COSY correlations were connected with key HMBC correlations. The aryl hydrocarbons of 3-H (δ_H_ 6.92 ppm) and 2-H (δ_H_ 6.71 ppm) exhibited HMBC correlations with the two aryl ketones 1-C and 4-C (δ_C_ 190.8 ppm and 184.9 ppm), respectively. HMBC correlations between the hydroxyl 5-OH (δ_H_ 12.49 ppm) and the quaternary aromatic carbon atoms (δ_C_ 115.5 ppm and δ_C_ 124.8 ppm) supported the intramolecular hydrogen bonding next to the aryl ketone carbon atom 4-C (δ_C_ 190.8 ppm). The last two aryl hydrocarbons 6-H and 7-H (δ_H_ 7.31 ppm and δ_H_ 7.24 ppm) showed HMBC correlations with the quaternary aromatic carbon atoms 5-C and 8-C (δ_C_ 161.9 ppm and δ_C_ 135.2 ppm) establishing these two aryl hydrocarbons next to the phenol carbon (δ_C_ 161.9 ppm) versus next to the aryl ketone carbons. Since the LC/MS analysis indicted a molecular mass of 346 g/mol, the NMR spectra of **1** suggested that the quinone is a symmetrical dimer, each half containing a hydrogen bonded hydroxyl and two pairs of *o*-coupled aryl hydrocarbons. Thus, the connectivity of the two halves must be either a 6,6′- or 8,8′-linkage. Based on the HMBC correlations, δ_H_ 7.24 (H-7,7′) and 6.71 ppm (H-2,2′) with δ_C_ 128.3 ppm (C-8a,8a’) as well as δ_H_ 12.49 (OH-5,5′), 7.31 (H-6,6′) and 6.92 ppm (H-3,3′) with δ_C_ 115.5 ppm (C-4a,4a’), the 8,8′-linkage was suggested. Previous work by H. Laatsch showed differences in the UV absorption maxima for synthetic 8,8′-bijuglone (437 nm) versus 6,6′-bijuglone (445 nm) as well as a larger downfield shifts of the OH-signal from monomer (δ_H_ 10.83) to the dimer of ∆δ_H_ of 0.7 ppm for 8,8′-linkage as observed here [26,27]. The 8,8′-dimeric naphthoquinone represents a unique case where the proton and carbon environments are chemically and magnetically equivalent. Other fungal 1,4-naphthoquinone dimers are known that demonstrate chemical and magnetically equivalent environments in NMR experiments, including maritinone and mamegakinone [28]. Biosynthetically, biaryl natural products can be constructed from an oxidative coupling of polyketide originated monomers catalyzed by a laccase, peroxidase, or cytochrome P450 enzymes (CYP) [29]. In plants, however, most axially connected naphthoquinones are derived from the shikimate pathway, as shown for maritinone and mamegakinone, isolated from persimmon fruit tree *Diospyros maritima* [30,31]. Naphthoquinone biosynthesis is common in microorganism and plants and can hint to either parallel chemical evolution or gene transfer events. For many plant metabolites, the producing organism might be found in endophytes [32]. Following the established structure of **1**, we wanted to determine if the fungus produced and retained **1** in its cells or if the compound was secreted. Extracts derived from fungal cells and supernatant of a liquid culture were analyzed by LC/MS. The [M-H]^-^ ion (extract-ion chromatogram at 345 *m/z*) was only observed in the cell extract and not in the supernatant, suggesting that **1** is kept intracellularly and not secreted into the medium.

### 2.2. Genome Analysis of Z. pseudotsugae

A 43.4 Mb draft genome for *Z. pseudotsugae* was submitted to the genome mining software antiSMASH to generate and identify BGCs (Figure 1) [33]. A total of 36 BGCs were predicted, including 17 type 1 polyketide synthases (TIPKS), 14 non-ribosomal peptide synthetase-like (NRPS-like), two terpenes, one ribosomally synthesized peptide (fungal-RiPP) and two hybrid pathways. Of these, a few BGCs showed a match for published small molecules when searched against the Minimum Information of the Biosynthetic Gene cluster (MIBiG) database against characterized gene clusters [34]. Some matches exhibited 100% similarity to known compounds, including (-)-mellein (T1PKS), cercosporin (NRPS), aureobasidin A1 (NRPS), phomopsins (fungal-RiPP), and elsinochrome A (T1PKS) (Figure 1). Interestingly, cercosporin and elsinochrome, resemble key features of 8,8′-bijuglone (**1**) (highlighted in blue in Figure 1) suggesting that **1** could possibly be a precursor or shunt metabolite related to these biosynthetic pathways. Cercosporin and elsinochrome A are both light-dependent, toxic pigments from fungi. Perylenequinones are believed to play key roles as virulence factors in the chemical interactions between plant-pathogenic fungi and their hosts. Both metabolites are able to absorb light energy and can produce reactive oxygen species (ROS). A high amount of these toxins is reported to induce necrotic lesions on citrus and tobacco leaves [35,36,37]. 8,8′-bijuglone might exhibit similar functions in planta.

### 2.3. Antimicrobial Activity

The compound **1** was evaluated for antimicrobial activity against three Gram-positive bacteria, methicillin-resistant *Staphylococcus aureus* (MRSA) BAA-41, *Bacillus subtilis* (ATCC 49343), and *Mycobacterium smegmatis* (ATCC 14468), two Gram-negative bacteria, *Escherichia coli* (ATCC 8739) and *Pseudomonas aeruginosa* (ATCC 15442), and a fungal human pathogen *Candida albicans* (ATCC 90027) (Table 2). Since quinones like *p*-quinones and 1,4-naphthoquinones are known to be unstable in DMSO stock solutions due to their apparent redox sensitivity [38,39,40,41], compound **1** was solubilized in ethanol for antimicrobial activity tests. Against MRSA and *B. subtilis*, treatments with **1** at 125 μg/mL [360 µM] were moderately antimicrobial, yielding in 29.3% and 32.6% bacterial cell survival respectively, compared to 100% cell survival for vehicle control wells (Table 2). Other compounds similar to **1** have been reported to have antibacterial activity, including 8,8′-biplumbaign (also called maritinone) [30,42], chitranone [30] and diospyrin [43].

### 2.4. Cytotoxicity Activity

The fungal extract of *Z. pseudotsugae* was tested against the human colon carcinoma cell line HCT-116 (ATCC^®^ CCL-247^TM^) in a single dose MTT based cell viability assay. The extract showed potent activity with 16% cell survival when tested at 10 µg/mL. Since the extract was highly enriched with **1**, we next purified 8,8′-bijuglone and established the IC_50_ value against HCT-116 to be 130 µM (0.13 mM) [45 µg/mL], exhibiting only weak cytotoxicity (Figure 2). Compound **1** was submitted to the National Cancer Institute (NCI) for evaluation against their NCI-60 cancer cell line panel [44]. The compound showed selective, lethal activity against all six leukemia cancer cell lines when tested at 10 µM (Figure 3). In order to obtain an IC_50_ value against a panel of leukemia cancer cell lines, Dr. Tom O’Hare and his team at Huntsman Cancer Institute, Utah tested the compound against eight acute myeloid leukemia (AML) cell lines different from the ones used by the NCI. Here, purified 8,8′-bijuglone had no effect against the tested cell lines, most likely due to its redox sensitivity and limited solubility in ethanol when tested from a 10 mM stock solution. We cannot rule out that a highly potent compound in minute quantities is causing the activity in the extract. However, compound content on the extract was assessed by measuring the area under the curve of the UV absorbance at 210 nm, which exhibited 84% 8,8′-bijuglone with the remaining 20% being derived from solvent and media peaks, and 90% purity was detected at 254 and 280 nm. No other metabolites were detected using light scattering detection nor via the mass detector, but this does not rule out another bioactive metabolite. Noteworthy, fungal cultures extracted with ethyl acetate (pH = 5.5) and kept neat retained potent cytotoxic and antimicrobial activity over time.

## 3. Materials and Methods

### 3.1. General Experimental Procedures

Optical rotation was determined on a JACS P-1010 polarimeter. IR spectra were recorded on a Thermo Scientific Nicolet IR100 FTIR spectrometer. LRESIMS were recorded on Agilent 1100 series LC with MSD 1946 (LC/MS). NMR spectra were measured on a Bruker Avance III 500 MHz spectrometer, equipped with a 5 mm TXI probe, with the residual solvent used as an internal standard (CDCl_3_: *δ*_H_ 7.26, *δ*_C_ 77.06) [45]. Semi-preparative HPLC was performed with an Agilent 1100 Infinity HPLC system equipped with photodiode array detectors. Semi-preparative HPLC and analysis by LRESIMS used a Phenomenex Kinetex C18 column (150 mm × 4.6 mm, 5 μm). All solvents (analytical grade, LCMS grade, and HPLC grade) were from Sigma-Aldrich Corp., Fisher Scientific, and VWR International.

### 3.2. Foliage Sampling

At each site, foliage was collected from second- and third-year internodes on secondary branches in the upper crowns of five randomly selected 10- to 30-year-old Douglas-fir trees. From one of the five trees sampled at each of the Swiss needle cast (SNC) sites, foliage samples were also collected from the lower, mid, and upper crowns to assess within-tree diversity. The foliage was stored on ice and promptly returned to the campus of Oregon State University for storage in a cold room for no longer than 5 days prior to processing. Needles with pseudothecia were attached to the lids of Petri dishes with double-sided adhesive tape, placed over water agar, and incubated for 48–72 h. Individual ascospores were removed from the agar with sterilized forceps and transferred onto 2% malt agar (MA) (Difco Laboratories, Detroit, MI). Cultures were incubated at 18° C for a minimum of 2–6 months.

### 3.3. Culture Media

An amount of 2% Malt: malt extract (2% g/L) (Criterion^TM^ Malt Extract, Hardy Diagnostic, Santa Maria, CA, USA) with no pH adjustment prior to sterilization. For agar plates 15 g/L nutrient agar was added to the culture media before sterilization. *Zasmidium pseudotsugae* was cultivated on 2% malt agar plates (MA) (40 × 25 mL) with a 1 cm^2^ piece of agar from a single culture and allowed to grow at ambient light and temperature for 5 months. *Zasmidium pseudotsugae* did not grow on rice solid media. Broth cultures (50 mL) were inoculated with a 1 cm^2^ piece of agar from the 5-month-old agar culture and allowed to grow at 28 °C on an orbital shaker at 110 rpm for 1 month. One-liter broth cultures were inoculated with 20 mL culture material from a 50 mL broth culture and allowed to grow at 28 °C on an orbital shaker at 110 rpm for 3 months.

### 3.4. Preparation of Organic Extracts and Vacuum Liquid Chromatography

The agar cultures were blended with an equivalent portion of ethyl acetate, while the broth cultures were extracted using equal parts ethyl acetate for 24 h with stirring. The organic layer, from either the agar cultures or broth culture, was concentrated under reduced vacuum.

### 3.5. Isolation and Physiochemical Properties of 8,8′-Bijuglone (***1***)

The ethyl acetate extract from the agar plate was subjected to semipreparative HPLC using an isocratic elution of (ACN/H_2_O (50:50) + 0.05% formic acid) for purification to yield 1.2 mg of compound **1**.

*8,8′-bijuglone* (**1**): orange, needle-like crystal; [α]_D_^20^ = 0; IR (DCM) ~3200, 2924, 2854, 1640; ^1^H-NMR (500 MHz, CDCl_3_) and ^13^C-NMR (500 MHz, CDCl_3_) see Table 1. 1D- and 2D- NMR spectroscopic data see Appendix A. The molecular formula was determined as C_20_H_10_O_6_ based on the QTOF-HRMS with *m/z* 346.0477 [M]^−^ calcd. for C_20_H_10_O_6_^−^ 346.04829 (0.1 ppm); IR (ATIR) 2924, 2854, 1640 cm^−1^.

### 3.6. Genome Assembly and Bio-Informatics Analysis

Draft genome assembly of *Z. pseudotsugae* BR-8-S5 isolated from a single Douglas-fir needle (Blue river reservoir, OR; 44.157615N, 122.323W) was obtained by generating paired-end Illumina reads at the Centre d’expertise et de services Génome Québec plateform, Montreal, QC. DNA was extracted from a pure 4-week-old culture (2% Malt extract + 0.2% yeast extract) using the Qiagen DNeasy Plant Mini kit. Genomic library was constructed according to the NEBnext Ultra II DNA library preparation kit for Illumina (version 3.2) with minor modifications and sequenced on Illumina HiseqX machine. Raw sequencing reads were submitted to quality control and assembled into contigs using ABySS [46] with a range of k-mer values from 32 to 96. The best assembly was then selected based on genome size and contiguity (best N50). This de novo assembly is available in the NCBI database under Bioproject number PRJNA599411. Prediction of specialized metabolite genes clusters was established by submitting de novo assembly contigs to the antiSMASH fungal version with default parameters [33].

### 3.7. Antimicrobial Assays

Organic extracts and isolated compound were tested for activity in cell-based microbroth single dose assays following established protocols [18]. The antimicrobial activity was evaluated against methicillin-resistant *Staphylococcus aureus* (BAA-44), *Bacillus subtilis* (ATCC 49343), *Mycobacterium smegmatis* (ATCC 14468), *Escherichia coli* (ATCC 8739), *Pseudomonas aeruginosa* (ATCC 15442) and *Candida albicans* (ATCC 90027). Antibiotic positive controls (vancomycin, chloramphenicol, rifampicin, ampicillin, kanamycin, and amphotericin, respectively) were used at 125 µg/mL [360 µM], while DMSO or ethanol was used as negative control at 1.25% *v*/*v*. Compound (**1**) was prepared at 10 mg/mL in DMSO or ethanol, added to wells in duplicate at a final concentration of 125 μg/mL [360 µM].

### 3.8. Cytotoxicity Assay

Cytotoxic activity of organic extracts and pure compound were evaluated against a human colon carcinoma model (HCT-116, ATCC CCL-247) in cell-based assays following established protocols [47]. Effects on mammalian cell viability were determined by measuring the reduction of the tetrazolium salt MTT (3-(4,5-dimethylthiazolyl-2)-2,5-diohenyltetrazolium bromide) by metabolically active cells. HCT-116 cells were maintained in MEM supplemented with 10% (*v*/*v*) fetal bovine serum, penicillin (100 U/mL), and streptomycin (100 *µ*g/mL). The cell lines were incubated at 37 °C in 5% CO_2_. Cells were plated into 96-well plates at 7000 cells/well cell density, incubated overnight, and treated with the addition of 10 *µ*g/mL compound and controls to each well. After 48 h, MTT (5 mg/mL in phosphate-buffered saline) was added to each well at a final concentration of 0.5 mg/mL. The plates were incubated for 2 h at 37 °C. The medium was removed, and the purple formazan product solubilized by the addition of 50 µL of DMSO. Absorbance was measured at 550 nm using the Biotek Synergy 96-well plate reader. Metabolic activity of vehicle-treated cells (0.1% *v/v* DMSO) was defined as 100% cell growth. Etoposide (250 µM) was used as a positive control. IC_50_ value for pure compound (**1**) was determined using a 10-point dilution dissolved in PBS + 5% ethanol.

Compound **1** was submitted to the NCI-60 panel and tested at a single dose of 10^−5^ M in ethanol [44]. One-dose data are reported as a mean graph of the percent growth of treated cells relative to the no drug control (Figure 3). Growth inhibition (values between 0 and 100) and lethality (values less than 0) can be extracted from the graph. **1** exhibited lethality within the leukemia panel. Effects of compound (**1**) on acute myeloid leukemia (AML) cells were determined at University of Utah [48]. The cytotoxicity activity was evaluated against eight AML cells: OCI-AML2, CMK, HL-60, MOLM-13, MOLM-14, KG1a, OCI-AML3, and SKM. Compound **1** was dosed up to 10 mM in ethanol, however no activity was observed at the highest concentration.

## 4. Conclusions

In the current study, a previously described 1,4-naphthoquinone dimer, 8,8′-bijuglone (**1**) was isolated for the first time from a natural source. The structure of **1** was elucidated by extensive spectroscopic analyses. **1** showed moderate antibacterial activity and potent cytotoxic activity when tested in a highly enriched extract and exhibited selective activity against leukemia cells when tested in the NCI-60 panel. However, when purified and tested against the human colon carcinoma cell line (HCT-116), only weak activity with an IC_50_ value of 130 µM [45 µg/mL] was observed. Naphthoquinone dimers are known to be redox sensitive with limited solubility. Current efforts are underway to stabilize the compound to allow for in-depth biological tests. Future studies aim to explore the mechanism of self-resistance of *Zasmidium* against 8,8′-bijuglone, as the compound was observed to be retained in the fungal cells, as well as the effects of **1** in planta. Additionally, genome analysis revealed a surprising large metabolic potential within *Zasmidium pseudotsugae*. To our knowledge, the current study presents the first secondary metabolite isolated from *Z. pseudotsugae* and is the first report of its biosynthetic potential using genome analysis approaches.

## Figures and Tables

**Figure 1 molecules-25-02358-f001:**
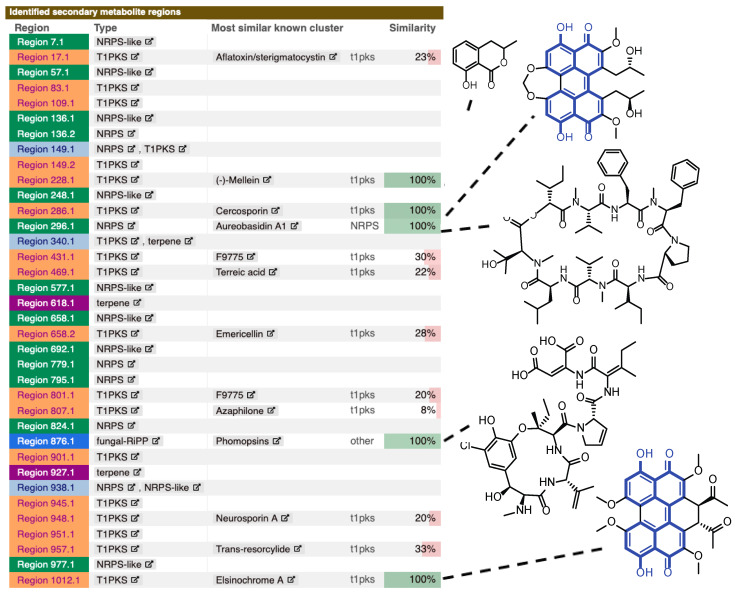
Biosynthetic gene clusters predicted in *Z. pseudotsugae* identified by antiSMASH. Clusters with 100% similarity to a compound are shown on the right. The structure of 8,8′-bijuglone (**1**) is highlighted in blue in the core structures of cercosporin and elsinochrome.

**Figure 2 molecules-25-02358-f002:**
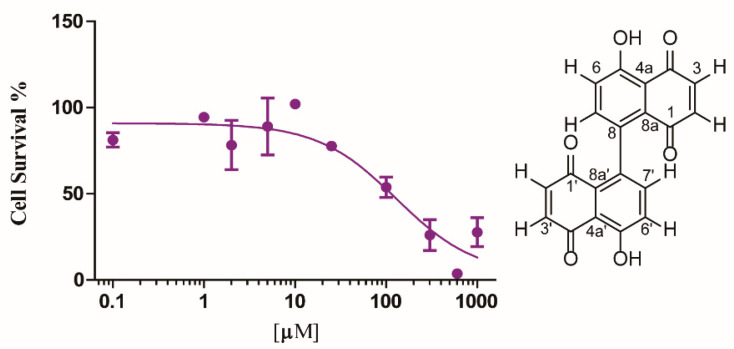
IC_50_ curve of 8,8′-bijuglone (**1**) against human colon carcinoma (HCT-116). The IC_50_ value was determined to be 130 µM (0.13 mM, 45 µg/mL).

**Figure 3 molecules-25-02358-f003:**
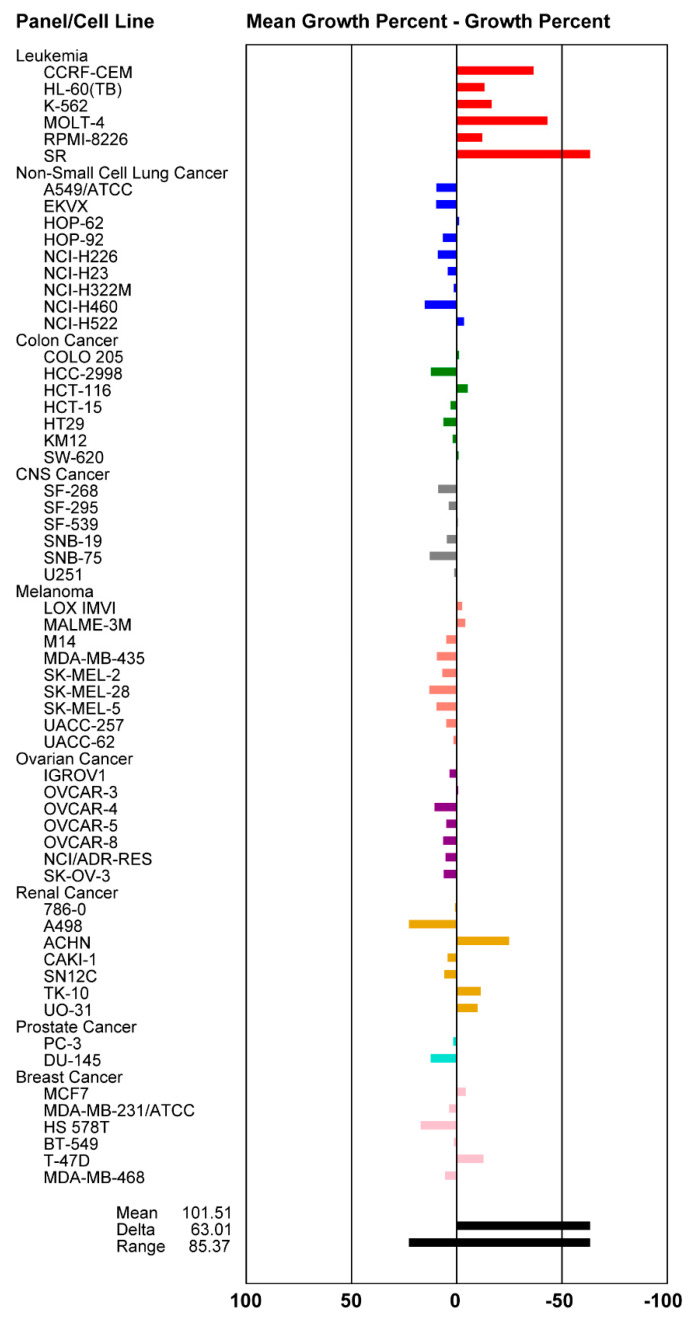
Mean graph display of NCI-60 cell line screening data for 8,8′-bijuglone (**1**) (NCS 811277). Sample concentration at 10 µM. The bars to the right indicate high lethality and the bars to the left indicate growth inhibition. Experiments were performed at the NCI-DTP: http://dtp.cancer.gov [44].

**Table 1 molecules-25-02358-t001:** ^1^H and ^13^C NMR of 8,8′-bijuglone (**1**) at 500 MHz in CDCl_3_.

	Compound		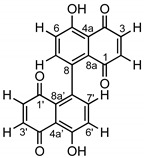
Position	δ_C_	Type	δ_H_, mult (*J* in Hz)
1, 1′	184.9	C	-
2, 2′	140.6	CH	6.71, d (10.2 Hz)
3, 3′	138.0	CH	6.92, d (10.2 Hz)
4, 4′	190.8	C	-
4a, 4a′	115.5	C	-
5, 5′	161.9	C	-
6, 6′	124.8	CH	7.31, d (8.7 Hz)
7, 7′	138.7	CH	7.24, d (8.7 Hz)
8, 8′	135.2	C	-
8a, 8a′	128.3	C	-
5-OH, 5′-OH	-	-	12.49, s

**Table 2 molecules-25-02358-t002:** Antimicrobial activity of 8,8′-bijuglone (**1**). Percent given is cell survival after treatment. Pathogens used: methicillin-resistant *Staphylococcus aureus* (BAA-44) *=* MRSA, *Bacillus subtilis* (ATCC 49343), *Mycobacterium smegmatis* (ATCC 14468), *Escherichia coli* (ATCC 8739), *Pseudomonas aeruginosa* (ATCC 15442), *Candida albicans* (ATCC 90027).

Sample ^1^			Antibacterial ^2^			Antifungal ^2^
	MRSA	*Bacillus subtilis*	*Mycobacterium smegmatis*	*Escherichia coli*	*Pseudomonas aeruginosa*	*Candida albicans*
8,8′-bijuglone	29.3%	32.6%	91.5%	97.1%	86.0%	99.5%
positive control	0.0% vancomycin	15.1% chloramphenicol	1.5% rifampicin	11.6% ampicillin	0.2% kanamycin	23.6% amphotericin B
negative control	> 100% ethanol	68% ethanol	100% DMSO	89.0% ethanol	92.4% ethanol	100% ethanol

^1^ Samples and positive controls were tested to a final concentration of 125 µg/mL [360 µM]. Negative controls were tested at 1.25% ethanol, except *M. smegmatis* in DMSO.

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
