# Peer review of "Chemical, Bioactivity, and Biosynthetic Screening of Epiphytic Fungus Zasmidium pseudotsugae"

_molecules, 2020, doi:10.3390/molecules25102358_

Round 1

Reviewer 1 Report

In this manuscript, the authors identified 8,8'-bijuglone from the plant associated fungi, Zasmidium pseudotsugae. The structural elucidation is well described. The authors claims that 8-8'-bijuglone has antibiotic activity agains gram bacteria and cytotoxicity. However, the cytotoxicity seemed not strong enough. Authors are saying that the cytotoxicity of 8-8'-bijuglone is low because of its instability. 

Measure point, 

The authors are testing the activity using crude extract and this crude extract seemed to have potent activity. However, this  activity is probably not from 8-8'-bijuglone. They say that the content of 8-8'-bijuglone in the extract is 80% but this result will not connect the activity with 8-8'-bijuglone siginificant enough. What happen if another compound in the extract (imagine it contained in the extract 10% to 5%) has nM order activity? They should revise the story or identify the compound which actually have potent cytotoxicity.

Minor points,

Please provide the HR-MS data.

Line 60, it is not so surprising that the fungi have 36 biosynthetic gene clusters. This is quite normal.

Table 1. I think 4a and 8a is preferred instead of 10 and 9.

Figure 1. This figure do not show the identification of the cluster. These are just predictions at this moment. 

Author Response

Thank you for your feedback, please find our point-by-point response below:

In this manuscript, the authors identified 8,8'-bijuglone from the plant associated fungi, Zasmidium pseudotsugae. The structural elucidation is well described. The authors claims that 8-8'-bijuglone has antibiotic activity agains gram bacteria and cytotoxicity. However, the cytotoxicity seemed not strong enough. Authors are saying that the cytotoxicity of 8-8'-bijuglone is low because of its instability. 

Measure point, 

The authors are testing the activity using crude extract and this crude extract seemed to have potent activity. However, this  activity is probably not from 8-8'-bijuglone. They say that the content of 8-8'-bijuglone in the extract is 80% but this result will not connect the activity with 8-8'-bijuglone siginificant enough. What happen if another compound in the extract (imagine it contained in the extract 10% to 5%) has nM order activity? They should revise the story or identify the compound which actually have potent cytotoxicity.

Thank you for your comment. Indeed, we cannot rule out that a highly potent compound in minute quantities is present, causing the activity. However, no other metabolites could be detected and de-replicated in the highly enriched extract from agar cultures via UV absorbance (DAD), ELSD, or mass spectrometry. The purity of 8,8’-bijuglone was determined by AUC to be at least 84% in the agar crude extract using the sensitive wavelength 210 nm, while at wavelengths 254 nm and 280 nm it was at 92% and 91% purity, respectively. The remaining 20% under 210 nm detection belong to signals from the injection time (polar) and column wash (unipolar). Since no signals were detected in the light-scattering detector nor the mass detector besides some common solvents and media components, with regards to our detection bias, we believe that no other small molecule was present. Overall, we have updated the manuscript to reflect the possibility.

Minor points,

Please provide the HR-MS data.

We have added the HRMS data to the SI. We were not able to receive the original data file due to the COVID pandemic, but show a screen shot from the direct injection. Our apologies.

Line 60, it is not so surprising that the fungi have 36 biosynthetic gene clusters. This is quite normal.

Yes, we agree. The statement relates to the fact that we only detected one metabolite in the fungal extract, we edited the text.

Table 1. I think 4a and 8a is preferred instead of 10 and 9.

Thank you, we have edited the text and tables accordingly.

Figure 1. This figure do not show the identification of the cluster. These are just predictions at this moment. 

Thank you, we have changed the text accordingly.

Reviewer 2 Report

The study "Chemical, bioactivity, and biosynthetic screening of epiphytic fungus Zasmidium pseudotsugae" authored by González-Montiel and collaborators is very well written and presented. The authors used a combined approach to isolate and characterize the compound 8,8’-bijuglone from a foliage-associated fungus. There is great interest in the potential of natural compounds in the development of novel antibiotics or anticancer drugs, and this manuscript should be compelling to the broad readership of Molecules. I support its publication in its current form, although I have three minor suggestions to the authors:

1) The Introduction is fine but could be enriched with another paragraph in the end, contextualizing some background on what is known about Naphthoquinones and antimicrobials and cytotoxicity, if anything. Or some other known function of these or similar compounds, just to bring non-specialists closer to your narrative.

2) Line 146, table 1, lines 257 and 259, the concentration of 125 ug/mL is used. In lines 162 and 296 it is referred to as molarity. It would be nice to include both concentration and molarity in all mentions during the antimicrobial and cytotoxicity assays so the reader more easily can relate these two aspects.

3) In Figure 3, consider reducing the scale to +100 to -100 so the bars will be taller and more easily differentiated.

Author Response

Thank you for your feedback, please find our point-by-point response below:

The study "Chemical, bioactivity, and biosynthetic screening of epiphytic fungus Zasmidium pseudotsugae" authored by González-Montiel and collaborators is very well written and presented. The authors used a combined approach to isolate and characterize the compound 8,8’-bijuglone from a foliage-associated fungus. There is great interest in the potential of natural compounds in the development of novel antibiotics or anticancer drugs, and this manuscript should be compelling to the broad readership of Molecules. I support its publication in its current form, although I have three minor suggestions to the authors:

  • The Introduction is fine but could be enriched with another paragraph in the end, contextualizing some background on what is known about Naphthoquinones and antimicrobials and cytotoxicity, if anything. Or some other known function of these or similar compounds, just to bring non-specialists closer to your narrative.

Thank you for your feedback. We have added some background in the genome analysis section on the biosynthesis of naphthoquinones as well as some of the previously observed functions.

  • Line 146, table 1, lines 257 and 259, the concentration of 125 ug/mL is used. In lines 162 and 296 it is referred to as molarity. It would be nice to include both concentration and molarity in all mentions during the antimicrobial and cytotoxicity assays so the reader more easily can relate these two aspects.

Thank you, we agree and added the molarity to antimicrobial assays and the ug/mL concentration to the cell viability assay.

  • In Figure 3, consider reducing the scale to +100 to -100 so the bars will be taller and more easily differentiated.

Thank you, we have removed some of the empty space and think the figure is much improved!

Reviewer 3 Report

This is an interesting and well written manuscript.

The main concern is whether authors checked if the compound was toxic to "normal" cells? in order to compare their cytoxicity data? Please advise.

Minor issues: figure 1 and 2 could be improved.

Author Response

Thank you for your feedback, please find our point-by-point response below.

This is an interesting and well written manuscript.

The main concern is whether authors checked if the compound was toxic to "normal" cells? in order to compare their cytoxicity data? Please advise.

Thank you for the comment, which addresses a rather complex matter. First, the assay we are reporting here – our-in house MTT based assay, the NCI-60 panel, and the AML panel, are all cell viability assays, they only measure cell metabolism, not toxicity. Assays for lethality would need to be done separately, and various ways are available from annexin-based FACS, to LDH release, to induction of apoptosis assays (via FACs, Western Blot, enzyme kits).

Second, no ‘normal’ cells are tested in the NCI-60 panel routine, and most immortal cell line that are available in chemistry labs are not considered normal either. While for some tissues, more ‘normal’ cells available, eg epithelia cells for melanoma work, these are often primary cells and not easy to work with.

For some general toxicity profile, we have used HEP2 liver cells line before or blood cells, however this was beyond the scope of this rather moderate activity profile.

Minor issues: figure 1 and 2 could be improved.

We have improved figure 2. Since there was no other comment about figure 1, we are not sure how to improve the figure and kept as is.

Round 2

Reviewer 1 Report

Authors edited the manuscript according to the comments by the referee. Now should be acceptable for the journal.